# The effect of natural selection on the propagation of protein expression noise to bacterial growth

**Laurens H. J. Krah** 🄸, **Rutger Hermsen** 🄸 *

Theoretical Biology and Bioinformatics, Department of Biology, University of Utrecht, Utrecht, The Netherlands

* r.hermsen@uu.nl

## Abstract

In bacterial cells, protein expression is a highly stochastic process. Gene expression noise moreover propagates through the cell and adds to fluctuations in the cellular growth rate. A common intuition is that, due to their relatively high noise amplitudes, proteins with a low mean expression level are the most important drivers of fluctuations in physiological variables. In this work, we challenge this intuition by considering the effect of natural selection on noise propagation. Mathematically, the contribution of each protein species to the noise in the growth rate depends on two factors: the noise amplitude of the protein's expression level, and the sensitivity of the growth rate to fluctuations in that protein's concentration. We argue that natural selection, while shaping mean abundances to increase the mean growth rate, also affects cellular sensitivities. In the limit in which cells grow optimally fast, the growth rate becomes most sensitive to fluctuations in highly abundant proteins. This causes abundant proteins to overall contribute strongly to the noise in the growth rate, despite their low noise levels. We further explore this result in an experimental data set of protein abundances, and test key assumptions in an evolving, stochastic toy model of cellular growth.

**Data Availability Statement:** All relevant data are within the manuscript and its Supporting information files.

**Funding:** LHJK was supported by the NWO (Nederlandse Organisatie voor Wetenschappelijk

## Author summary

Gene expression in bacterial cells is intrinsically stochastic: copy numbers of all proteins vary between genetically identical cells, even in a homogeneous environment. Such noise in gene expression affects metabolic fluxes and even propagates to the cellular level. Indeed, also the growth rate of individual cells, an important proxy for fitness, fluctuates strongly. This beckons the question as to which protein species contribute most to the noise in the cell's growth rate. We here derive general mathematical predictions stating that if cells have been optimised by evolution to grow fast, abundant protein species contribute most to the noise in the growth rate. This result is counter-intuitive, because the noise levels in the expression of those highly expressed proteins are small.

Onderzoek, www.nwo.nl) (Grant 022.005.023). The funders had no role in study design, data collection and analysis, decision to publish, or preparation of the manuscript.

**Competing interests:** The authors have declared that no competing interests exist.

## Introduction

Stochasticity is inherent to gene expression [1–3]. Stochastic variation in the copy numbers of proteins is observed even under constant external conditions, and among individual cells in a population of isogenic bacteria. How cellular stochasticity, and noise in protein expression specifically, interferes with the functioning, survival and fitness of bacteria has been of great interest for many years [3–8].

Noisy gene expression is indeed commonly accepted as the dominant mechanism behind the strong phenotypic variation that has been observed in populations of genetically identical cells [9]. In an exponentially growing population of cells, even the growth rate of individual cells is distributed surprisingly broadly [10]. Since cellular growth rate (and its population average) is often considered an important proxy for bacterial fitness, the growth rate—and how its variation is shaped by noisy gene expression—has received much attention [11–14]. Notably, noise in the concentration of metabolic proteins is shown to propagate from the protein-level, via the metabolic network, to the instantaneous single-cell growth rate [15].

Commonly, noise is characterised in terms of the coefficient of variation (CV), defined as the standard deviation divided by the mean. In snapshots of bacterial populations, proteins with a higher mean expression level ($\mathbb{E}[X]$) generally have a lower coefficient of variation squared ($CV^2$) [16, 17]. For proteins with a low mean expression, noise is dominated by the intrinsic stochasticity of the chemical reactions involved [18, 19] and $CV^2$ scales as $1/\mathbb{E}[X]$ [2, 5, 16, 20–22]. For higher mean expression, noise levels decrease to eventually reach a plateau, where fluctuations in gene expression are dominated by extrinsic noise, such as noise resulting from cell division or environmental noise. Because of their larger noise levels, lowly expressed proteins are commonly assumed to be particularly important drivers of fluctuations in variables at the cellular level, such as the growth rate. At the same time the effects of the relatively small fluctuations of highly abundant proteins have largely been neglected.

However, while protein noise levels are mainly determined by mean abundance, the mean abundance itself is a product of evolution. Under many external conditions, bacteria indeed seem to tune their protein levels in order to grow, on average, at a near-optimal rate [23–28]. Optimal gene expression for fast growth has also been an important and fruitful assumption in countless modelling studies and techniques concerning deterministic growth, including Flux Balance Analysis [29–31]. So far, however, the possible effects of natural selection on how noise in protein expression affects the noise in macroscopic variables such as the growth rate, have not been considered.

In this work, we therefore consider bacteria whose protein expression levels are shaped by natural selection acting on the population growth rate. For the extreme case of cells growing optimally fast, we obtain analytical predictions for the contribution of each protein to the noise in the growth rate as a function of its mean expression only. The main result, directly opposing common intuitions, is that proteins with a high mean expression are most important for the noise at the cellular level. The argument is, in short, that a protein's contribution to noise in the growth rate does not only depend on the protein's noise level, but also on the sensitivity of the growth rate to that protein's fluctuations. We show that when protein expression levels are optimised for fast growth, the growth rate becomes most sensitive to fluctuations of abundant protein species. This causes abundant proteins to overall contribute strongly to the noise in the growth rate, despite their low noise levels. In a stochastic toy model of gene expression and growth, we verify and further investigate the role of natural selection on shaping noise propagation properties. Lastly, an analysis on experimental data of protein abundances and protein noise levels indicates that the common intuition –cells behave noisily because of the low copy number of certain molecular species– might be incorrect.

## Results

To grow, bacteria need to express a certain set of (metabolic) proteins. Together, these proteins create a metabolic flux used to build cellular components and new proteins. Since the cell's growth rate is limited by this metabolic flux, noise in the expression levels of the proteins involved propagates through the metabolic network to affect the growth rate [15]. For any fixed external environment, we therefore assume the existence of an unknown function $\mu(\mathbf{X})$ that describes the instantaneous rate of cellular growth as a function of the copy numbers of all proteins, $\mathbf{X}$. The growth rate is thus a deterministic function of stochastic variables. In a population snapshot, different individuals stochastically express different copy numbers of their proteins and hence the growth rates of individuals will differ.

To quantify how variation in the expression of protein $i$ affects growth rate $\mu$, we use previously defined Growth Control Coefficients, which measure the sensitivity of the growth rate to small changes in the copy number of protein $i$ [32]:

$$C_i^\mu \equiv \left( \frac{X_i}{\mu} \frac{\partial \mu}{\partial X_i} \right) \Bigg|_{\mathbb{E}[\mathbf{X}]}. \tag{1}$$

Here, the expectation value is taken over the distribution of protein copy numbers across a population of cells. As we will show below, these GCCs offer a way to decompose and analyse the noise in the growth rate in terms of contributions by each of the noisy components, the proteins.

To arrive at a comprehensive and useful noise decomposition, we adhere to two simplifying assumptions. First, noise levels are assumed to be small, so that all protein abundances are close to their means. The growth rate can then be approximated as a linear function of the protein levels. Secondly, fluctuations in all protein species are assumed to be independent. In bacterial cells, this is certainly not the case. However in the case of correlated fluctuations, noise contributions can not be uniquely defined [33, 34]. Indeed, when two proteins correlate, and their joint fluctuations affect growth, the attribution of the noise contribution to either protein is arbitrary. Therefore, we here present the simplified case where all protein abundances are uncorrelated so that noise contributions can be uniquely defined and understood intuitively.

Under these assumptions noise in the growth rate, $CV_\mu^2$, can be approximated as a sum of contributions from all protein species:

$$CV_\mu^2 \approx \sum_i CV_i^2 (C_i^\mu)^2, \tag{2}$$

where $CV_i$ is the coefficient of variation of the copy number of protein species $i$ across a population of cells (see S1 Appendix for the derivation). In this equation, each protein's contribution consists of two factors. The first factor is no surprise: the proteins' coefficient of variation which quantifies the fluctuations in the expression of that particular protein. The second factor is the protein's GCC, which quantifies how strongly these fluctuations actually affect the cellular growth rate.

### Distribution of growth control coefficients

To further quantify which proteins are important for noise in the growth rate, we need to gain more insight into how growth control is distributed among proteins. This distribution is not arbitrary due to three properties of the GCCs, which are discussed below.

**Sum rule.**   Firstly, the sum of the GCCs equals zero [32]:

$$\sum_i C_i^{\mu} = 0. \tag{3}$$

This sum rule originates from the so-called *intensivity* of the growth rate: if all protein copy numbers inside a cell are increased by the same factor, the cellular flux increases, but the mass increases as well, such that the cell's growth rate (mass increase per mass) stays the same. That the growth rate is to a good approximation an intensive variable has been shown in multiple experiments [10, 15] and is a common modelling assumption [30]. Moreover, it is analogous to the assumption that the metabolic flux and cellular mass are *extensive* variables, which has been used in Metabolic Control Analysis to derive a similar sum rules for fluxes [35].

**H-proteins.**   Secondly, there is a set of proteins, here called *H*-proteins, that are crucial for the cell's survival, but do not contribute to metabolism or cellular growth. This set *H* includes 'house-keeping proteins' participating in, *e.g.*, stress-response, immunity, and DNA damage repair; in bio-engineering, *H* may also contain engineered pathways. In wild type *Escherichia coli*, the *H*-sector comprises an estimated 25–40% of the total protein mass [36]. Even when *H*-sector proteins are not toxic or otherwise harmful to the cell, their control on the growth rate will still be negative. This is because their synthesis does take up resources that otherwise could go to growth-related enzymes. Previously, the GCC of such *H*-sector proteins has been calculated [32] to be equal to their mass fraction:

$$C_{i \in H}^{\mu} = -\phi_i. \tag{4}$$

Here we write $\phi_i := \mathbb{E}[X_i]/\sum_j \mathbb{E}[X_j]$ for the proteome mass fractions $\phi_i$ of each protein species *i* and ignore, for simplicity of notation, that different proteins have different masses. The mass fraction of the total *H* sector is denoted as $\phi_H$.

Together with the sum rule, the presence of the *H*-sector has important consequences for the distribution of GCCs: because some proteins have negative GCCs, others must have a positive GCC.

**Optimal growth.**   Thirdly, natural selection tends to favour populations of cells that, on average, grow faster. This drives the (mean) expression levels of many proteins to be (near)-optimal for growth [24, 25]. We here show that this also affects their GCCs by considering the extreme case of a cell in which the expression levels of all proteins (except for those in the *H* sector) are fully optimised for growth.

To do so, evolution is treated mathematically as a constrained optimisation problem, where the mean growth rate, $\mathbb{E}[\mu]$, is optimised under two constraints. First, the cell's protein density is kept constant. Second, only a fixed fraction of the proteome $(1 - \phi_H)$ can be allocated towards proteins related to growth. To maximise the growth rate, only the (mean) protein abundances inside this fraction can be tuned by evolution while the total abundance must stay the same.

Formally, the optimisation can be done using Lagrange multipliers on a linearisation of $\mu$ (see S1 Appendix). For all proteins that are not in the *H* sector, the result is the following important expression for the GCCs in the optimal state:

$$C^{\mu*} = \left(\frac{\phi_H}{1 - \phi_H}\right) \phi_i^*. \tag{5}$$

Here, the asterisk indicates that the equation is only valid under optimality. Intuitively, the result can be understood as follows. In the optimal state, all partial derivatives of the growth rate must be equal: if the growth rate would increase more upon increasing $\mathbb{E}[X_i]$ than upon

increasing $\mathbb{E}[X_j]$, increasing the expression of $i$ at the expense of $j$ would increase the growth rate, and hence the growth rate would not be optimal [37, 38].

Eq 5 reveals two important properties for cells optimised for growth. First, growth control is shared between all metabolic proteins: there is no single growth-limiting protein. Secondly, and most importantly, enzymes with a higher mean expression level have a proportionally larger control on the growth rate.

## Combining all factors

The distribution of the GCCs in optimally growing cells (Eq 5) can be combined with the experimentally observed scaling of coefficients of variations to predict the contribution of each protein species $i$ to the noise in the growth rate. We write $\kappa_i^P$ for this contribution, which is defined as the protein's relative contribution to the CV of $\mu$ as expressed in Eq 2:

$$\kappa_i^P := \frac{(C_i^\mu)^2 \mathrm{CV}_i^2}{\mathrm{CV}_\mu^2} \approx \frac{(C_i^\mu)^2 \mathrm{CV}_i^2}{\sum_j (C_j^\mu)^2 \mathrm{CV}_j^2}. \tag{6}$$

Inspired by experimental data [16, 22], the intrinsic noise component is assumed to be inversely proportional to mean abundance:

$$\mathrm{CV}_i^2 = F/\mathbb{E}[X_i], \tag{7}$$

with a fixed Fano Factor $F$. If we ignore the noise plateau caused by extrinsic noise sources, Eq 7 sets the noise levels of all protein species. Note that for the highly expressed proteins, noise levels are thus deliberately underestimated, resulting in a conservative estimate for their contribution to noise in the growth rate.

To now analyse noise propagation in optimally growing cells, Eqs 4, 5 and 7 are inserted in Eq 6. This results in:

$$\kappa_i^{P^*} = \begin{cases} \dfrac{\phi_H}{1-\phi_H} \phi_i^* & \text{for } i \text{ not in } H, \\[2ex] \dfrac{1-\phi_H}{\phi_H} \phi_i & \text{for } i \text{ in } H. \end{cases} \tag{8}$$

This equation is the pivotal finding of this study. It states that, in cells whose expression levels are optimised for growth, $\kappa_i^P$ is proportional to $\phi_i$, that is, proteins with a high mean expression contribute most strongly to fluctuations in the growth rate.

The implications of the above equations become clear when applied to a data set of measured protein abundances and protein noise levels in the model bacterium *E. coli* [16, 39]. Under the assumptions of Eqs 2 and 5, the top 5% most abundant protein species are estimated to contribute over 90% of the noise in the growth rate (Fig 1, red dots). This contribution is significantly larger than might have been expected *a priori*: If all GCCs are assumed equal (Fig 1, purple dots) these abundant proteins contribute only 40%, and if the GCCs of the optimal state are shuffled so that the correlation between a protein's abundance and its GCC is broken (Fig 1, black dots) their contribution becomes negligible (<2%). On the other hand, the 50% least abundant proteins are estimated to contribute only 2% of the overall noise in the growth rate (Fig 1, red curve), instead of 50–90%. For details of the analysis, see S2 Appendix and S1 Fig.

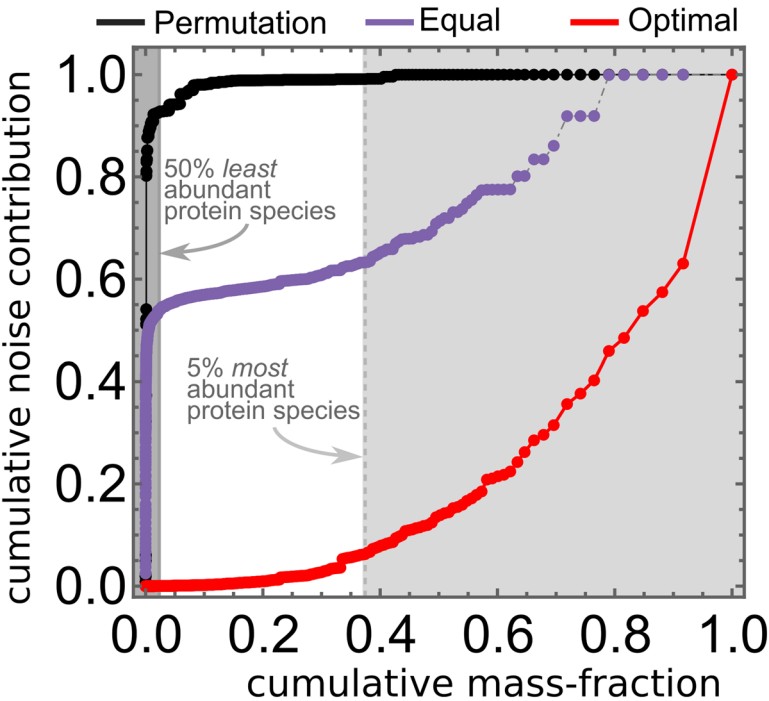

**Fig 1. Cumulative noise contribution as a function of cumulative mass fraction, both calculated from protein abundances and noise levels measured in *E. coli* [16].** Protein species were ordered by their mass fraction $\phi_i$, and the cumulative mass fractions and noise contributions subsequently calculated as $\sum_{j=1}^{i} \phi_j$ and $\sum_{j=1}^{i} \kappa_j$, respectively. GCCs are either set by Eq 5 (red dots), all equal (purple dots), or a random permutation of the optimal GCCs (black dots).

## Stochastic toy model

To examine if our results still hold for the large noise levels seen in living cells, we study noise propagation in a stochastic toy model of a growing, and evolving, cell. Specifically, we test if a positive scaling between $\phi$ and $\kappa$ can still be observed for genotypes that have evolved by random mutations, under realistic noise levels.

To do so, we defined a highly simplified model of a growing cell with stochastic protein expression levels. To mimic the effects of evolution, we then employ random mutations to search for the mean protein expression levels that optimise the mean growth rate of such cells. Next, we characterised the noise propagation in such optimised cells to verify the predictions of Eqs 5 and 8.

The model cell consists of a linear metabolic pathway consisting of five reactions that import an external metabolite ($m_1$) and convert it to biomass (Fig 2A). Each reaction is catalysed by a single enzyme species and inhibited by its own product. Additionally, a sixth protein species is expressed that is not metabolically active, representing the *H*-sector. Given the abundances **X** of all proteins, the instantaneous growth rate $\mu$ is defined as the steady state flux through the pathway divided by the total number of expressed proteins, including the *H*-sector. Note that the growth rate depends non-linearly on all protein abundances. (For more details, see Methods).

The abundances **X** themselves are stochastic: each $X_i$ is distributed in the population according to a Gamma distribution [16, 20] characterised by mean $\mathbb{E}[X_i]$ and a Fano factor $F$. The Fano factor is chosen the same for all proteins, consistent with Eq 7, and sets the overall noise amplitude in the cell.

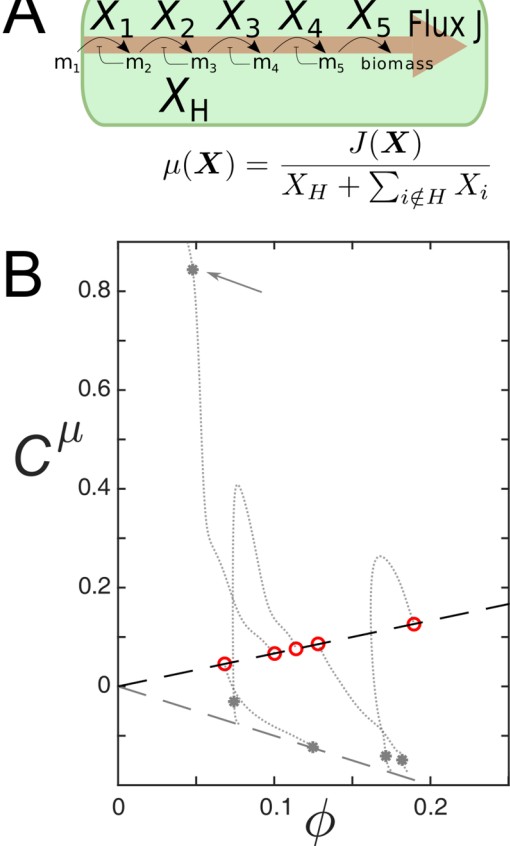

**Fig 2.** *(A)* Representation of the stochastic toy model. A cell expresses five metabolic protein species, each catalysing a single reaction in a linear reaction chain that imports and converts a fixed external metabolite, $m_1$, into biomass. The growth rate is defined as the steady state flux through the network, divided by the total number of expressed proteins, including the $H$-sector protein. *(B)* Example trajectory of the GCCs of metabolic proteins during the optimisation of a single kinotype (grey dotted lines). Grey dots are GCCs at an early stage of this process and the arrow indicates the rate-limiting protein species. Red point are the values of the GCCs in the optimal genotype, matching the predicted scaling for metabolic proteins (positive dashed line, negative dashed line is the prediction for $H$-proteins). See S1 Table for parameters.

It is useful to distinguish three levels of description of the model cells: their kinotype, genotype, and phenotype. We introduce the 'kinotype' as the set of reaction parameters that fully characterise the enzymes in a cell's metabolic network: kinetic rates, Michaelis–Menten constants and inhibition parameters of the five reactions. We define a cell's 'genotype' as the mean abundances of all protein species. Lastly, a cell's 'phenotype' is given by the vector of the current protein abundances and the corresponding growth rate. The phenotype is therefore a multi-dimensional stochastic variable whose probability distribution depends on the genotype.

During an evolutionary trajectory, the genotype is repeatedly subjected to mutations that are subsequently either rejected or accepted. Mutations increase or decrease the mean abundance of one particular protein species, after which the mean expression of all other metabolic proteins is adjusted such that the total (mean) protein abundance remains fixed ($\sum_i \mathbb{E}[X_i] = \Omega = 10^4$ in all simulations, for more details see Methods). Mutations thus affect the protein copy number distributions across the population and therewith also the probability distribution of the growth rate. A mutated genotype is accepted only if it increases the

population mean growth rate, which is determined by sampling many phenotypes generated by that genotype. The evolutionary algorithm is halted when 100 consecutive mutations around a particular genotype are rejected. The resulting genotype is expected to be close to a local optimum, although it is not guaranteed to be exactly the mathematical optimum due to sampling error.

Using a method adopted from [33] the noise contribution of each protein species is then measured by again sampling and analysing many phenotypes. These measured noise contributions were compared with our prediction, $\kappa^P$ (Eq 8). Lastly, the whole process above was repeated for many kinotypes (randomly generated; see Methods), resulting in different optimal genotypes.

**Low-noise regime.** Before analysing the more realistic regime of high noise levels (large variance of the protein copy numbers), we first study the model in a low-noise regime (using $F = 1$, $\Omega = 10^4$, resulting in small copy-number variance), where our results (Eqs 2 and 8) are expected to hold well. When noise levels are this low, the mean growth rate $\mathbb{E}[\mu]$ is well approximated by the growth rate in the vector of mean abundances, $\mu(\mathbb{E}[\mathbf{X}])$. Instead of using the undirected, slow stochastic evolutionary algorithm described above, the growth rate was therefore optimised using a deterministic gradient-based hill climb algorithm (for details see Methods).

During each step of the optimisation process, we measured the GCCs of the metabolic proteins to observe how they adjust during optimisation. A representative example of such a trajectory is shown in Fig 2B (dotted grey lines). Early in the optimisation process, when genotypes are still far from optimal (grey dots), often one particular protein species (shown in the figure with an arrow) is strongly limiting growth ($C_i^\mu \approx 1$, indicating that the growth rate could be improved by increasing this protein's expression level). In contrast, the expression of other metabolic proteins is too high; those proteins have a negative GCC similar to $H$-sector proteins, indicating that almost all of their expression is a burden to the cell. Eventually, as fitter genotypes are found, growth control becomes shared among all proteins (Fig 2B, dotted grey lines). When the optimisation algorithm simulation has found the optimal genotype, the predicted positive scaling between a protein's GCC and its mean abundance is obtained (Fig 2B, red points, and Eq 5).

Repeating the same process for multiple kinotypes ($n = 10$) confirms the generality of the positive scaling between $C^\mu$ and $\phi$ after optimisation (Fig 3A, red points). Again, note that in an early stage of the optimisation process the distribution of the GCCs is markedly different: Although only metabolic proteins are shown in the figure, some of them have negative GCCs that resemble the GCCs of $H$-sector proteins (Fig 3A, grey dots).

Next, we measured for each kinotype the noise contributions of each protein in the optimal genotype, and, for comparison, in a non-optimal genotype. For all kinotypes, the noise contributions in the optimal genotypes neatly follow our prediction (Fig 3B, red points). In contrast, for non-optimal genotypes, noise contributions are dominated by only a few lowly expressed proteins (Fig 3B, grey dots). The mean abundance of these proteins is below the optimal value, causing both their GCC and their CV to be large, resulting in a large noise contribution.

This analysis clearly highlights the fundamental role of evolution in shaping noise propagation properties: only in evolved cells that grow at an (almost) optimal rate a positive scaling exist between $\kappa$ and $\phi$.

**High-noise regime.** Next we study the toy model in a high-noise regime, where copy number variation matches observed variation in living bacteria more closely ($F = 10$, $\Omega = 10^4$, resulting in CVs up to 0.2, S5(C) Fig). In this regime, the non-linear dependence of the growth rate on the protein abundances becomes important and might influence the mean growth rate. That is, a genotype that was optimal in the low-noise regime, does not necessarily yield the

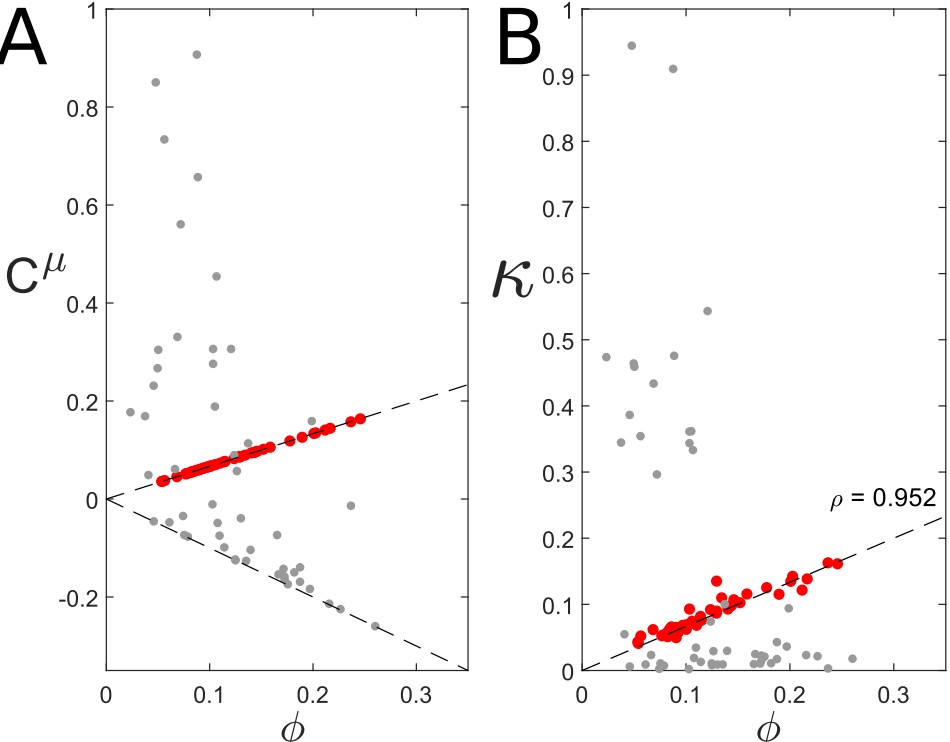

**Fig 3. Perfect prediction in the case of linear noise for 10 different kinotypes.** *(A)* Grey dots are GCCs of metabolic proteins in genotypes after 150 optimisation steps, which are not yet optimal. Red points are the GCCs in the optimal genotypes. Dashed lines are predictions for the values of GCCs for metabolic (positive) or *H*-proteins (negative). *(B)* Measured noise contributions compared to predicted noise contributions (dashed line). Red points are measured in the optimal genotype, grey points in the non-optimal genotypes.

highest mean growth rate in the high-noise regime. Below, we therefore distinguish two genotypes for each kinotype: the 'Low-Noise' (LN) genotype, which is optimal in the low-noise regime, and the 'High-Noise' (HN) genotype, which evolved in the high-noise regime by random mutations. To efficiently find the HN genotype of each kinotype, we employed the evolutionary algorithm described above, starting with the LN genotypes.

For some kinotypes, the resulting HN genotypes indeed differed significantly from the LN ones. This can be understood as follows. In the LN genotype, when noise levels are low, control over the growth rate is shared between the proteins. In the same LN genotype, higher noise levels can increase the probability that in some sampled phenotypes a single protein species becomes the sole rate-limiting step. Indeed, some genotypes that were optimal in the low-noise regime generated many slow-growing phenotypes when noise levels were high (S2(B) Fig). The low growth rates were often caused by a single protein species whose phenotypic expression was too low (S2(A) Fig) and therefore became a bottle-neck.

From an allocation point of view, increasing the mean expression of lowly abundant, rate-limiting proteins is cheap: little additional resources are needed to cause a relatively large change in the protein's expression level. Indeed, the mean expression of potential bottlenecks increased during evolution, but mainly for lowly expressed protein species (S2 and S3 Figs).

The distribution of the GCCs is also different in the evolved HN genotypes compared to the LN genotypes (Fig 4A, red points). GCCs are by definition linear measures and because in the high-noise regime the non-linearity of the growth rate become relevant, Eq 5 is not expected to hold exactly anymore. Interestingly, however, the positive scaling between $\phi$ and $C^\mu$

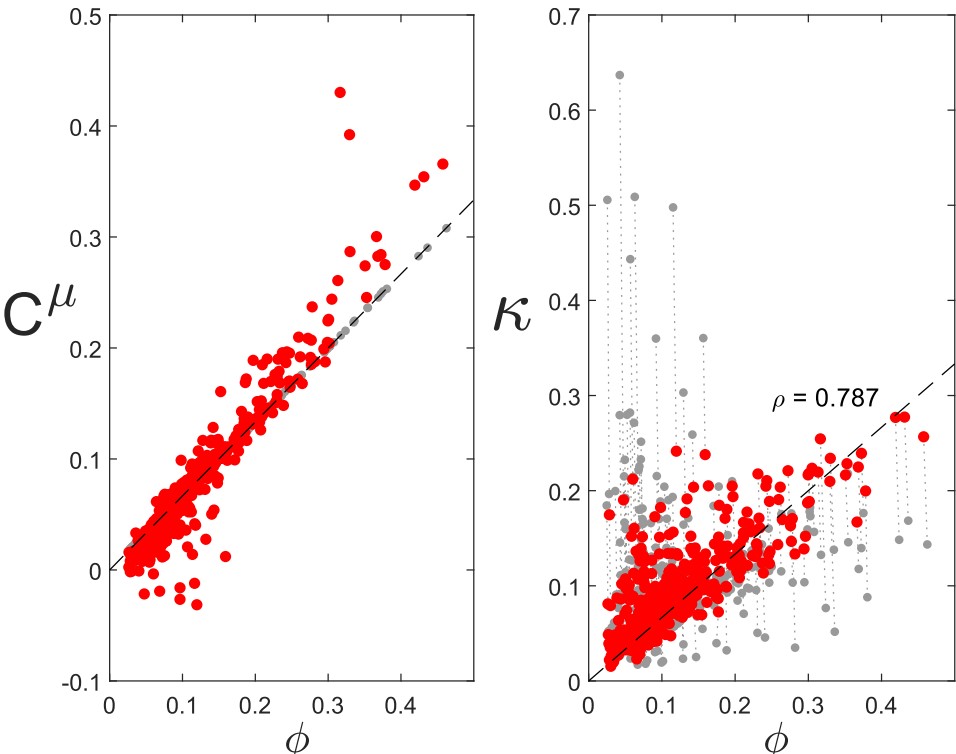

**Fig 4. For 100 kinotypes, low-noise genotypes (grey dots) compared with high-noise genotypes (red dots).** Dashed lines are theoretical predictions for metabolic proteins. *(A)* Growth Control Coefficients. *(B)* Measured noise contributions.

remains, and becomes, if anything, even steeper. Again, this makes sense: when noise levels increase, lowly expressed proteins are, due to their larger CV, more likely to fluctuate down to levels that strongly limit growth (S3(A) Fig). Increasing the mean expression of these proteins will reduce their GCC (S3(B) and S3(C) Fig), while increasing the GCC of the other proteins via the sum rule (Eq 3). The net result is an increase in the slope in Fig 4A.

Importantly, the positive scaling between $\kappa$ and a protein's mean expression was again observed in the high-noise regime (Fig 4B, red points). This is remarkable, since the mathematical prediction (Eq 8) was derived under the assumption of small noise amplitudes. To understand why the positive scaling nonetheless persists in the high-noise regime, the same reasoning as for the GCCs can be followed. At the start of the evolutionary trajectory, when the genotype is still the LN genotype, some lowly abundant protein species are dominant noise contributors (Fig 4B, grey points top-left corner). During evolution those proteins obtained a slightly higher expression, reducing both their GCC and their noise contribution $\kappa$ (Fig 4B, red points, and S3(B) and S3(C) Fig), and at the same time increasing the contributions of the other proteins species.

## Discussion

In this article we argued that highly expressed proteins play an important role in system-level noise properties despite their low noise levels. In summary, our argument is that, in cells optimised for growth, abundant protein species have large Growth Control Coefficients (GCCs). The product of a protein's CV and GCC –an indicator for the protein's noise contribution– is then predicted to be proportional to the protein's mean abundance. A crude estimate from an

*E. coli* data set suggests that the 50% least abundant protein species contribute only $\sim$2% of the noise in the growth rate, instead of up to 90% as might have been expected *a priori* if the scaling of the GCCs is ignored. In contrast, the top 5% most abundant species are estimated to contribute well over 80% in optimised cells. In a simplistic toy model of cellular growth, we then showed that the positive scaling between a protein's mean abundance and its noise contributions persist even when noise levels are considerable and the growth rate is a non-linear function of protein abundances.

Of course, these results rely on the assumption of a perfect optimum. Experiments suggest, however, that bacteria are not always in such an optimum [24]. In certain fixed environments, adaptive mutants arise with higher growth rates, indicating that the wild-type growth rate is not optimal yet [40]. Assuming those adaptive mutations did not take place in the *H*-sector, it remains a question whether in wild-type cells abundant proteins also contribute most to noise in the growth rate. However, in our simulations the positive scaling between $\phi$ and $\kappa$ persists even in the high noise regime, where genotypes were evolved by random mutations–and therefore are not necessarily perfectly optimal. Moreover, some mutants generated around the evolved genotype still displayed the positive scaling even when growing, on average, significantly slower (S6 Fig). Other generated mutants, however, did lose their positive scaling. It would therefore be interesting to further investigate why some non-optimal mutants still display the predicted scaling, but others do not.

While the toy model assumed particular enzyme dynamics, all mathematical predictions were derived without any assumptions concerning the underlying biochemistry. This implies that the strong contribution of highly expressed proteins to the noise in the growth rate is a general property of evolved biological systems as long as protein expression is in some way constrained.

The presence of such cellular constraints is crucial for our results. In this study, a tight constraint was imposed by assuming that the *H*-sector is completely static. However, a small relaxation of this constraint -*e.g.*, assuming that the allocation towards the *H*-sector has to be within a certain range- should yield similar results. Also other allocation constraints, *e.g.* a fixed total protein abundance allocated to a particular metabolic pathway, or a maximum density of membrane proteins [41], will result in a similar positive scaling between $\phi$ and $\kappa$ in all constrained proteins.

We point out that, besides the growth rate, other cellular traits, such as stress response or antibiotic resistance, are important for bacterial fitness as well. Interestingly, noise in these traits can be analysed in the same way as noise in the cellular growth rate. Therefore, the argument suggests that noise in any intensive trait that has been optimised during the bacteria's evolutionary history should be dominated by highly expressed proteins.

Throughout this paper, we ignored all correlations between protein species [42, 43], because in the presence of such correlations, the contribution of noise in a particular protein to the noise in cellular growth rate becomes ill defined. One way to circumvent this problem is to use a fine-grained model description, *e.g.* at the level of individual chemical reactions, in which the noise sources are inherently uncorrelated [6, 44], or to adopt a meta-modelling approach [34] where highly correlated protein species (*e.g.*, those coded in the same operon) are modelled as single, noise contributing units. That said, the method presented here allows for a more intuitive interpretation of noise contributions, because it directly relates noise contributions to observed protein abundances.

The results discussed above add to the realisation that global cellular constraints have intricate consequences for the overall physiology of evolved cells, from noisy gene expression [32], to metabolism [31, 41] and growth [45]. Our work highlights the holistic nature of noise propagation via the sum rule for the GCCs (Eq 3). The sum rule specifically could have important

consequences for biotechnology: tinkering with a specific part of the cell affects noise propagation properties of the entire system. For example, most synthetic proteins or pathways do not contribute to growth, but instead create by-products. Such pathways will thus have a negative GCC and hence increase the GCC -and therewith the propagation- of all metabolic proteins.

The stochastic toy model moreover revealed a trade-off between efficient resource allocation and the robustness of metabolism to expression noise [46]. Genotypes that are optimal in the low-noise regime allocate resources efficiently, but lack metabolic robustness at higher noise levels. Cells with different levels of expression noise therefore required different genotypes to grow, on average, the fastest (Fig 4). Similar observations were also made in recent experiments in yeast [14]. Together, these observations can have consequences for Flux Balance Analysis-like techniques [29], where optimal growth states have so far been calculated mostly deterministically, *i.e.*, optimising the growth rate in the mean expression levels.

We conclude that noise in gene expression –and its propagation towards the growth rate– needs to be considered when discussing optimal growth, but also *vice versa*: when enzyme expression is optimised, this affects noise propagation in such a way that abundant protein species become most relevant for noise on a system level.

## Methods

Here, we give detailed information about the toy model and the stochastic simulation.

### Kinotype, genotype, phenotype and growth rate

In our toy model, we simulated a linear chain of 5 proteins, where the flux through each of the first 4 steps is given by:

$$v_i = \frac{k_{\mathrm{cat},i}\, X_i\, m_i}{k_{\mathrm{m},i} + m_i + m_{i+1}/k_{\mathrm{inh},i+1}}. \tag{9}$$

Here, $k_{\mathrm{cat}}$ is the reaction rate, $k_{\mathrm{m}}$ is the Michaelis-Menten constant, and $k_{\mathrm{inh}}$ is the inhibition constant, together called a kinotype. To define a kinotype, we sample $k_{\mathrm{cat}}$ and $k_{\mathrm{m}}$ uniformly from the interval [0.1, 6.1] and $k_{\mathrm{inh}}$ from [1, 7]. The external metabolite, $m_1$ is set to 10 and kept constant, simulating a fixed environment. The fifth protein in the reaction chain creates biomass and is not inhibited.

In a snapshot of a population of cells, the values of $\mathbf{X}$ are distributed according to some probability distribution $P(\mathbf{X})$, here assumed to be six independent Gamma distributions. Because the variance in the expression of each protein species is tied to its mean (Eq 7), the Gamma distribution for each protein species is completely specified by its mean copy number. Together, the mean copy numbers are called a genotype and fully define the probability distribution $P(\mathbf{X})$. According to $P(\mathbf{X})$, we can sample for each individual in a population a vector $\mathbf{X}$. This vector is referred to as the individual's phenotype. Corresponding with the phenotype, an individual's growth rate can be calculated by integrating the system of ODEs for all metabolites (for $i \in \{2, 3, 4, 5\}$, $\dot{m}_i = v_{i-1} - v_i = 0$ with each $v_i$ specified in Eq 9) until steady state ($10^6$ time steps in Matlab 2019b). The dynamics of the metabolites are assumed fast, such that they are always in steady state relative to the sampled protein copy numbers. Moreover, stochasticity in the metabolic reactions is ignored for the same reason: metabolic fluctuations relax on timescales much faster than growth (seconds rather than minutes or hours).

The specific kinetics chosen (Eq 9) then enforce that for all possible phenotypes a steady state flux exists. In the steady state, where all fluxes are equal, we define $v_i = v_{i-1} = J$, and:

$$\mu := J/\left(\sum_i X_i\right).$$

## Evolution

To simulate evolution, we start with an initial genotype and measure its mean growth rate (by sampling many phenotypes and calculating the corresponding growth rates), and subsequently mutate the genotype to search for genotypes that result in higher mean growth rates.

The initial genotype is constructed by sampling 5 uniform numbers, $\mathbf{x} \in [0, 1]^5$ and setting $\mathbb{E}[\mathbf{X}]_{\text{initial}} = \mathbf{x}\phi_H/\sum_j x_j \Omega$, with $\phi_H = 0.4$ (the proteomic fraction allocated to the $H$-sector) and $\Omega = 10^4$, the (mean) total protein abundance, here assumed to be equal to cell size.

Genotypes were mutated in two different ways, depending on the noise amplitude chosen (value of $F$) in the simulation. In the low noise regime ($F = 1$), the next genotype was determined with a gradient-based hill climbing algorithm that uses the current GCCs:

$$\mathbf{X}_{t+1} = \mathbf{X}_t + \epsilon\, \delta\mathbf{X_t}/|\delta\mathbf{X_t}|^2, \quad \delta\mathbf{X_t} = \frac{\Omega}{5}\left(\mathbf{C}^{\mu}_t/\phi_t - \sum_{j\notin H} C^{\mu}_{j,t}/\phi_{j,t}\right),$$

where $\epsilon$ is small (0.0002). This algorithm changes the genotype in the direction of the steepest growth rate increase.

In the high-noise regime ($F = 10$), changes in the genotype are due to random mutations, where one of the metabolic proteins is chosen at random and its mean abundance is changed according to a percentage drawn from a normal distribution (mean zero, variance 5%), after which the entire genotype is renormalised to enforce a fixed mean cell size of $\Omega$. A mutant genotype is accepted only if it yields a higher mean growth rate (calculated over $2 \cdot 10^4$ sampled phenotypes). The evolutionary process is terminated when 100 mutants have been rejected. The full evolutionary process is repeated 15 times; from these 15 evolutionary trajectories, the genotype with the highest mean growth rate is chosen to be the HN genotype. (The number 15 is arbitrary, but deemed enough to ensure the simulation did not get stuck in a local optimum, while still being computationally feasible).

## Calculating noise contributions

For a specific genotype, we measured noise contributions as follows. First, we sampled $2 \cdot 10^4$ phenotypes and calculated the corresponding growth rates. Then, we divided the distribution for each protein in 100 bins, and calculated the mean growth rate in each bin (sampling extra if less than 100 phenotypes fell in a particular bin). Afterwards, we calculated the weighted variance between these growth rates. This is an approximation of a conceptual decomposition method from Bowsher and Swain [33]:

$$\kappa_i = \frac{\text{Var}[\mathbb{E}[\mu|X_i]]}{\text{Var}[\mu]}. \tag{10}$$

This method is a first order approximation of a full Global Sensitivity Analysis [34]. The approximation is only valid if the sum of all contributions is close to unity. This is indeed the case (see S5(A) Fig). Moreover, sampling errors in $\kappa$ are small (S5(B) Fig). All Matlab and Mathematica codes are available upon request.

## Supporting information

**S1 Appendix. Derivation of Eqs 2–8.**
(PDF)

**S2 Appendix. Application to experimental data sets.**
(PDF)

**S1 Fig. Examination of the effect of the noise floor and application to a second experimental data set.** *(A)* Data from Taniguchi *et al.*, but variances were now assumed to solely scale with mean protein abundance ($CV^2 = 1/\mathbb{E}[X]$), ignoring the noise floor ($n_f = 0$, see S2 Appendix, black dots in the figure). *(B)* Effect of adding the noise floor on predicted noise contributions in an optimally growing *E. coli* cell. Adding a noise floor (in this case using measured variances) increases the noise contribution of a few very abundant protein species (their noise levels increase), but also causes many low copy number protein species to contribute relatively less. *(C)* Cumulative noise contributions ($\Sigma\kappa$) against cumulative mass fractions ($\Sigma\phi$) estimated from the Schmidt *et al.* data set. GCCs were again set according to optimal growth (red dots), equal (purple dots), or shuffled (black dots). Shaded areas indicate the 50% least abundant protein species (left) and the top 5% most abundant species (right).
(EPS)

**S2 Fig. Example of highly skewed distributions in an optimised kinotype.** *(A)* Distribution of the efficiency of the fifth protein ($\eta_i := \frac{k_{\text{cat},i}}{\mu}\frac{X_i}{\sum_j X_j}$). *(B)* Distribution of growth rates for the optimal Low-Noise genotype in the high noise regime. Red areas in (A) and (B) correspond to the same phenotypes. *(C)* The same as (A), but for the evolved High-Noise genotype. *(D)* the same as (B), but for the evolved, High-Noise genotype. Grey distribution is the distribution in the LN-genotype for comparison. (We picked this kinotype because it most clearly showed the effect of evolution on the distributions).
(EPS)

**S3 Fig. Relation between a protein's efficiency and change in growth control.** *(A)* Probability a protein's efficiency is very close to unity, calculated over $2 \cdot 10^4$ phenotypes. Although a higher efficiency on first glance seems good, an efficiency close to unity indicates that this protein might be limiting growth. All proteins for with $p(\eta > 0.99) > 0$ are marked with a red cross. Two outliers (triangles) are protein species from the same kinotype. *(B)* Changes in genotype when noise levels increase, as a function of the mean $\phi$ in the optimal genotype. Red crosses are those proteins marked in panel A. Note that the highly expressed proteins with $p(\eta > 0.99) > 0$ are not increased, probably because this required the allocation of too much additional resource. *(C)* Changes in mean genotype coincide with a change in $C^\mu$.
(EPS)

**S4 Fig. Example of the optimisation algorithm.** *(A)* Growth rate increases each step due to the gradient-based hill climb algorithm. *(B)* Correlation coefficient increases and switches sign during optimisations. *(C)* Variance decreases during most parts of the optimisation process. *(D)* $CV_\mu^2$ decreases. Parameters of this example kinotype are equal to Fig 1B.
(EPS)

**S5 Fig. Examination of the noise decomposition method and noise levels.** *(A)* The sum of all first order noise contributions is close to unity, indicating that a first order Global Sensitivity Analysis captures the variance contributions well. *(B)* Noise contributions for the example kinotype in the LN genotype in the high noise regime. Error bars indicate 2 sd over 125

repeated sampling of $2 \cdot 10^4$ phenotypes. Sampling errors in $\kappa$ are within reasonable bounds. *(C)* Distribution of CVs as encountered in HN genotypes. (EPS)

**S6 Fig. Examination of the sensitivity of the positive scaling.** $\phi$ versus $\kappa$ for kinotype 2 around the evolved genotype (black circle) in the high noise regime ($F = 10$). Grey dots denote mutant genotypes, for which their mean growth rate (relative to the mean growth rate of the evolved genotype) is plotted against the correlation coefficients of $\phi$ and $\kappa$ for that particular genotype. Mutants are created by three different methods: (1) mutation in a single protein with a normally distributed step size with standard deviation 15% (40 genotypes, plus-sign markers), (2) with 5% (30 genotypes, diamond-sign markers), and (3) mutations in all protein species with standard deviation 5% (30 genotypes, square-sign markers). After the mutations, genotypes were re-normalised to make sure total cell size, and $\phi_H$ remained constant. (EPS)

**S1 Table. Kinetic parameters of the example kinotype.** Catabolic rate ($k_{cat}$) and inhibition constant ($k_{inhi}$) for the five metabolic protein species in the kinotype used to create figure Figs [2] and [S4]. Additionally, the initially sampled (relative) protein abundances are given. (PDF)

## Author Contributions

**Conceptualization:** Laurens H. J. Krah, Rutger Hermsen.

**Formal analysis:** Laurens H. J. Krah, Rutger Hermsen.

**Supervision:** Rutger Hermsen.

**Visualization:** Laurens H. J. Krah.

**Writing – original draft:** Laurens H. J. Krah.

**Writing – review & editing:** Laurens H. J. Krah, Rutger Hermsen.

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
