## [Decision Letter · Decision Letter 0]

29 Mar 2021

Dear Mr. Krah,

Thank you very much for submitting your manuscript "The effect of natural selection on propagation of protein expression noise to bacterial growth" for consideration at PLOS Computational Biology.

As with all papers reviewed by the journal, your manuscript was reviewed by members of the editorial board and by several independent reviewers. In light of the reviews (below this email), we would like to invite the resubmission of a significantly-revised version that takes into account the reviewers' comments.

We cannot make any decision about publication until we have seen the revised manuscript and your response to the reviewers' comments. Your revised manuscript is also likely to be sent to reviewers for further evaluation.

Sincerely,

Christopher Rao

Associate Editor

PLOS Computational Biology

Jason Haugh

Deputy Editor

PLOS Computational Biology

Reviewer's Responses to Questions

**Comments to the Authors:**

Reviewer #1: A well written and interesting manuscript. Some specific comments:

L 70: It might be worst stating explicitly that the expectation is defined over a distribution for X (which, in turn, is defined in terms of a population of cells)

L 161: please include a bit more detail for the derivation of equation 7. I’m not reaching this conclusion by combining 4,5, and 6 (and using the definition on line 117). It strikes me that we should get different results depending on whether i is in H or not (in which case I presume this is the "i not in H" case.

Figure 2A: are there meant to be two dashed lines in this figure? and red dots along the H-protein line; are some of the dots H-proteins?

L326: suggestion: can the function of operons be understood in this context? (i.e. not just coordinated expression but also, to an extent, resulting concerted variance in abundance)? Perhaps this case can be treated trivially by lumping all enzymes coded by an operon into a single ‘meta-protein’ to which you analysis applies directly. Just a thought.

L 351: It may be other referencing here the work by Terry Hwa’s group on protein partitioning and it’s relation to growth rate and other environmental factors, e.g.. Klumpp, Stefan, and Terence Hwa. "Bacterial growth: global effects on gene expression, growth feedback and proteome partition." Current opinion in biotechnology 28 (2014): 96-102.

Appendix S1: cell mass is described in terms of X (protein only). There’s I suppose, an unstated assumption that other contributors to cell mass (lips, nucleic acids), are proportional? (protein constituting maybe about 55% of the cell mass http://book.bionumbers.org/what-is-the-macromolecular-composition-of-the-cell/#:~:text=So%20the%20lipid%20bilayer%20occupies,this%20value%20to%20%E2%89%8814%25.)

Appendix S2: ’the flux through *each of* the first four steps is given by…

Omega can perhaps be defined more specifically as mean total protein abundance.

Reviewer #2: I enjoyed reading this work. I think the main result is of interest to a general audience, however little effort is made to make the text understandable, and of interest, to a more general audience than the biophysicist, familiar with key single-cell noise results. Also how significant this effect really is, when one consider for instance realistic E coli numbers and key abundant enzymes, such as the ribosome or glyceraldehyde-3-phosphate dehydrogenase, is unclear and would help the reader in appreciating the importance of the result in this work. I also miss a discussion of the relevance of protein copy number vs concentrations, isn't concentration noise the relevant measure for this purpose? Finally, noise in abundant protein is likely very small, in particular when noise in concentrations is considered, but how small? And whether the effect emphasised by the authors is truly significant in realistic situations remains unclear and deserves more attention for this work to be of interest to a more general audience than biophysicists developing novel theories -- I see this more as an intermediate result on the way to something more profound than a final result.

Reviewer #3: The manuscript by Krah and Hermsen investigates the propagation of protein noise to noise in bacterial growth. This is an interesting and timely manuscript. The authors built on their earlier work (ref 19) that introduces a sum rule for growth control coefficients and show that in cells that have optimised their growth (through natural selection), counter intuitively, the highly expressed genes (that have lower intrinsic noise) have the largest contribution to the growth rate variability. The authors show this in an idealised case, analytically and through simulation of a specific stochastic toy model. I have the following specific comments on the paper.

- The toy model is not fully stochastic. They assume expressions are static and follow a gamma distribution, but they are fixed and obtain the growth rate by solving systems of ODEs. Why did the authors not consider a fully stochastic model (using Gillespie simulations, for example).

- I am also confused with the so-called mutations and low and high noise regime, in their toy model. I believe they are simply considering two situations, one where growth is optimal (using a numerical optimisation of concentrations of the proteins, in the low noise case) and one where growth is not optimal and parameters are random (the high noise case, although, they still say “a mutant genotype is accepted if it yields a higher growth rate”. They also use some arbitrary number of repeats (15). I am very confused about the significance of the specific choices and the generality of the results. I find invoking the idea of mutations a bit confusing, could the authors not explore two situations (one where X’s are optimal for growth and one where X’s are not optimal for growth).

- I have a related point of confusion; the authors look at the effect of noise in expression in the optimal growth state on variability of growth rate. However, gene expression variability could have detrimental effect on growth rate itself (for example variability in expression of an essential low expressed enzyme could halt growth). Again, as the authors do not consider a fully stochastic model, they cannot address this issue.

- The authors do not consider extrinsic noise, but they should explain what happens if a constant base line as extrinsic noise is considered in their analysis.

**Have all data underlying the figures and results presented in the manuscript been provided?**

Reviewer #1: Yes

Reviewer #2: Yes

Reviewer #3: Yes

PLOS authors have the option to publish the peer review history of their article (what does this mean?). If published, this will include your full peer review and any attached files.

Reviewer #1: No

Reviewer #2: No

Reviewer #3: No
---

## [Decision Letter · Decision Letter 1]

22 Jun 2021

Dear Mr. Krah,

We are pleased to inform you that your manuscript 'The effect of natural selection on the propagation of protein

expression noise to bacterial growth' has been provisionally accepted for publication in PLOS Computational Biology.

Best regards,

Christopher Rao

Associate Editor

PLOS Computational Biology

Jason Haugh

Deputy Editor

PLOS Computational Biology

Reviewer's Responses to Questions

**Comments to the Authors:**

Reviewer #1: My concerns have been addressed.

Reviewer #3: I am happy with the revisions and I would like to recommend publication of the revised paper.

**Have the authors made all data and (if applicable) computational code underlying the findings in their manuscript fully available?**

Reviewer #1: None

Reviewer #3: None

PLOS authors have the option to publish the peer review history of their article (what does this mean?). If published, this will include your full peer review and any attached files.

Reviewer #1: No

Reviewer #3: No

---

## [Editor Report · Acceptance letter]

13 Jul 2021

PCOMPBIOL-D-20-02311R1 

The effect of natural selection on the propagation of protein
expression noise to bacterial growth

Dear Dr Krah,

I am pleased to inform you that your manuscript has been formally accepted for publication in PLOS Computational Biology. Your manuscript is now with our production department and you will be notified of the publication date in due course.

With kind regards,

Olena Szabo
